# Identification and Copy Number Variant Analysis of Enhancer Regions of Genes Causing Spinocerebellar Ataxia

**DOI:** 10.3390/ijms252011205

**Published:** 2024-10-18

**Authors:** Fatemeh Ghorbani, Eddy N. de Boer, Michiel R. Fokkens, Jelkje de Boer-Bergsma, Corien C. Verschuuren-Bemelmans, Elles Wierenga, Hamidreza Kasaei, Daan Noordermeer, Dineke S. Verbeek, Helga Westers, Cleo C. van Diemen

**Affiliations:** 1Department of Genetics, University Medical Center Groningen, University of Groningen, 9713 GZ Groningen, The Netherlandsh.westers@umcg.nl (H.W.); 2Department of Artificial Intelligence, University of Groningen, 9700 AK Groningen, The Netherlands; 3Commissariat à l’Énergie Atomique et aux Énergies Alternatives (CEA), Centre National de la Recherche Scientifique (CNRS), Institute for Integrative Biology of the Cell (I2BC), Université Paris-Saclay, 91198 Gif-sur-Yvette, France

**Keywords:** spinocerebellar ataxia, enhancer, 4C-seq, genetic diagnosis

## Abstract

Currently, routine diagnostics for spinocerebellar ataxia (SCA) look for polyQ repeat expansions and conventional variations affecting the proteins encoded by known SCA genes. However, ~40% of the patients still remain without a genetic diagnosis after routine tests. Increasing evidence suggests that variations in the enhancer regions of genes involved in neurodegenerative disorders can also cause disease. Since the enhancers of SCA genes are not yet known, it remains to be determined whether variations in these regions are a cause of SCA. In this pilot project, we aimed to identify the enhancers of the SCA genes *ATXN1*, *ATXN3*, *TBP* and *ITPR1* in the human cerebellum using 4C-seq, publicly available datasets, reciprocal 4C-seq, and luciferase assays. We then screened these enhancers for copy number variants (CNVs) in a cohort of genetically undiagnosed SCA patients. We identified two active enhancers for each of the four SCA genes. CNV analysis did not reveal any CNVs in the enhancers of the four SCA genes in the genetically undiagnosed SCA patients. However, in one patient, we noted a CNV deletion with an unknown clinical significance near one of the *ITPR1* enhancers. These results not only reveal elements involved in SCA gene regulation but can also lead to the discovery of novel SCA-causing genetic variants. As enhancer variations are being increasingly recognized as a cause of brain disorders, screening the enhancers of *ATXN1*, *ATXN3*, *TBP* and *ITPR1* for variations other than CNVs and identifying and screening enhancers of other SCA genes might elucidate the genetic cause in undiagnosed patients.

## 1. Introduction

The spinocerebellar ataxias (SCAs) are a heterogeneous group of autosomal dominantly transmitted neurodegenerative disorders that arise from progressive degeneration of the cerebellum and brainstem [1]. SCAs are clinically characterized by loss of balance and coordination, leading to severe disability. In the clinic, SCA patients are genetically diagnosed based on the presence of polyQ repeat expansions and conventional variations within protein-coding and the splice regions of known SCA genes, but this still leaves a group of SCA patients without a genetic diagnosis. The presence of known variants within non-coding regulatory regions (5′UTR, promoter and 3′UTR) of SCA genes causing SCA8, SCA12, and SCA37 [2,3,4] demonstrates that genetic variations that affect gene regulation can also play an important role in SCA pathogenesis. Enhancers are non-coding regulatory elements that can activate the transcription of genes in a tissue-specific manner. Increasing evidence shows that variations in enhancers, which can be located at up to several hundred kilobases from their target genes, can cause neurodevelopmental and neurodegenerative disorders such as intellectual disability, Alzheimer’s disease, Parkinson’s disease, schizophrenia, and Rett-like syndrome [5,6,7,8,9]. This led us to hypothesize that enhancers of SCA genes may also harbor variants that cause SCAs. However, the enhancers of SCA genes and their role in disease pathogenesis have not yet been studied.

Identifying gene enhancers is challenging due to their tissue specificity, the large distances between enhancers and their target genes, and the fact that multiple enhancers can regulate a single gene [10]. While the PubMed and enhancer databases include ~1.6 million putative enhancers specifically identified in the human brain, their target genes are mostly unknown due to the limited data on enhancer–promoter interactions [11,12]. Thus, it is still necessary to experimentally identify and validate gene-specific enhancers.

Given the high number of SCA genes and the heavy workload required to identify and validate gene-specific enhancers, we carried out a pilot study to identify enhancers of four SCA genes—*ATXN1*, *ATXN3*, *TBP* and *ITPR1*—in the human adult cerebellum. These four genes are either causal for the most common SCA types or dosage-sensitive. Studies have shown that increases in wild-type ATXN1 level, a protein involved in transcription regulation, induce SCA1 [13]. SCA3 is the most common type of SCA caused by polyQ tract in ATXN3, a deubiquitinating enzyme [14]. Furthermore, the loss of wild-type TBP, a general transcription factor, is the cause of SCA17 [15,16], and *ITPR1*, a calcium channel, is a dosage-sensitive gene for which haploinsufficiency is known to cause SCA15 [17]. Identifying cerebellar-specific enhancers of SCA genes can expand our knowledge of the tissue-specific molecular mechanisms governing the regulation of SCA genes and provide new avenues for genetic diagnostics.

## 2. Results

### 2.1. Identifying Enhancers of ATXN1, ATXN3, TBP and ITPR1 in Human Cerebellum

To identify the enhancers of the four SCA genes, we performed 4C-seq on a human cerebellum to capture genomic regions that interact with the promoter viewpoints. The 4C-contact peaks (Appendix A) were then overlapped with publicly available datasets to identify PutEs (Putative Enhancers), and we performed reciprocal 4C-seq for the PutEs to confirm their interaction with the corresponding promoter (Reciprocal-PutEs (R-PutEs)) (Appendix A). Finally, we performed luciferase assays for the R-PutEs to identify the active enhancers of each gene. We present detailed descriptions of the results for each gene below. PutEs, R-PutEs, and active enhancers identified for each gene can be found in Appendix A.

***ATXN1* enhancer identification.** 4C-seq data analysis for the *ATXN1* promoter showed six significant 4C-contact peaks (Appendix A). Based on the overlap of the 4C-contact peaks with open chromatin and H3K27ac data, we identified two PutE regions for *ATXN1*: PutE1 and PutE2 (Figure 1A). The *ATXN1* promoter and PutE2 were located in one TAD, whereas PutE1 was located in the neighboring TAD (Appendix A). An inter-TAD interaction between the two neighboring TADs [18] in which the *ATXN1* promoter and PutE1 are located can explain the interaction between them. Both PutEs were confirmed to interact with the promoter through reciprocal 4C-seq (R-PutE1 and R-PutE2) (Appendix A) and showed an increase in luciferase activity (2.4- and 7.2-fold, respectively) (Figure 2). R-PutE1 is located in the last intron of *ATXN1.* R-PutE2 is located in an intergenic region upstream of *ATXN1*. Based on the reciprocal 4C-seq results, no significant mutual interactions were observed between the R-PutEs and PutEs of *ATXN1*.

***ATXN3* enhancer identification.** 4C-seq data analysis for the promoter of *ATXN3* identified eight significant 4C-contact peaks (Appendix A) from which five PutEs were identified: PutE1–PutE5 (Figure 1B). The *ATXN3* promoter and PutE1–PutE5 were all located in one TAD (Appendix A). Reciprocal 4C-seq experiments for these PutEs confirmed that three of them, R-PutE1, R-PutE2, and R-PutE4, interacted with the promoter (Appendix A), but only R-PutE1 and R-PutE2 showed an increase in luciferase activity (59.4- and 5.9-fold, respectively) (Figure 2). R-PutE1 is located in the first intron of *PPP4R3A*. R-PutE2 is located in the first exon and intron of *TRIP11.* Reciprocal 4C-seq experiments did not show mutual interactions between the R-PutEs and PutEs of *ATXN3*.

***TBP* enhancer identification.** 4C-seq data analysis for the *TBP* promoter identified three significant 4C-contact peaks (Appendix A) from which two PutE regions were identified: PutE1 and PutE2. Both are located in the same TAD as the *TBP* promoter (Figure 1C and Appendix A). Reciprocal 4C-seq experiments confirmed that both PutEs interacted with the *TBP* promoter (R-PutE1 and R-PutE2) (Appendix A). Both R-PutE1 and R-PutE2 showed an increase in luciferase activity (6.7- and 27.1-fold, respectively) (Figure 2). R-PutE1 encompasses *DYNLT2*, while R-PutE2 contains two long intergenic non-coding RNAs: *LINC00242* and *LINC00574*. Reciprocal 4C-seq also revealed a mutual interaction between R-PutE1 and R-PutE2 (Appendix A).

***ITPR1* enhancer identification.** 4C-seq data analysis for the *ITPR1* promoter showed 11 significant 4C-contact peaks (Appendix A), of which seven were considered PutEs (Figure 1D). Three PutEs (PutE1, PutE2 and PutE3) and the *ITPR1* promoter are within one TAD. The other four PutEs (PutE4–PutE7) are located in a neighboring TAD (Appendix A). However, as seen in Figure 1D, a looping across the two neighboring TADs (an inter-TAD) [18] can explain the interaction between the promoter and PutE4–PutE7. Reciprocal 4C-seq experiments confirmed only the interaction of PutE2 and PutE7 with the promoter (R-PutE2 and R-PutE7) (Appendix A). R-PutE2 showed a 23.9-fold increase in luciferase activity. For cloning purposes, R-PutE7 was analyzed as two fragments (A and B) (Appendix A). While fragment A showed a 7.3-fold increase in luciferase activity, fragment B showed no increase (Figure 2). R-PutE2 encompasses intron 8 of *SUMF1* and exon 1 and intron 1 of *SETMAR*. Fragment A of R-PutE7 is located in intron 1 of the ncRNA *BHLHE40-AS1*.

Reciprocal 4C-seq analysis showed significant mutual interactions between the PutEs and R-PutEs of *ITPR1* (Appendix A), including the following: PutE1 with R-PutE2, PutE4 with R-PutE7, PutE4 with PutE6, PutE5 with PutE6, PutE5 with R-PutE7, and PutE6 with R-PutE7 (Figure 3A,B). Based on this, we decided to check the enhancer activity for PutE1, PutE4, PutE5, and PutE6 (Figure 3C). Due to their size, PutE1 and PutE5 were both analyzed as two separate fragments (A and B). Both PutE1 fragments showed an increase (9.5- and 4.6-fold, respectively) in luciferase activity, but only fragment B of PutE5 showed an increase (6.5-fold). PutE4 did not show any enhanced luciferase activity, whereas PutE6 showed an increase (6.9-fold) (Figure 3C). Although PutE1, PutE5, and PutE6 are active enhancer regions, they cannot be considered as enhancers for *ITPR1* because we could not confirm their interaction with the *ITPR1* promoter.

### 2.2. CNV Analysis

We next screened the enhancers of the *ATXN1*, *ATXN3*, *TBP*, and *ITPR1* of CNVs in a cohort of genetically undiagnosed SCA patients. CNV analysis did not reveal deletions or duplications in these regions in any of the patients. However, in one patient, we noted a CNV deletion in one of the 4C-contact peaks of the *ITPR1* promoter that did not overlap with the delineated active enhancer in this region (PutE1) (Figure 4). This CNV deletion had a minimum size of 40,590 bp (20 SNP probes, GRCh37; chr3:3,799,881–3,840,471). Examination of the SNPs flanking the deleted region allowed us to delimit the deletion to 3836 bp on the centromeric side (between rs711544 and rs139693033) and 294 bp on the telomeric side (between rs711582 and rs11705993). To identify the breakpoints of the deletion within the delimited regions, we designed several sets of primers at different locations and performed Sanger sequencing on the PCR products. The deleted region was 41,104 bp (UCSC; GRCh37; chr3:3,797,792–3,838,896). It encompasses part of intron 11 of the old version of the annotated *SUMF1* transcript (NM_182760) on the telomeric side and an intergenic region upstream of this transcript on the centromeric side, but it does not encompass the currently used *SUMF1* transcript (NM_182760.4).

Although the deletion did not overlap with PutE1, as the array data from the in-house control population did not show deletions affecting this locus, we still decided to perform follow-up studies for this deletion. The luciferase assay for the deleted region of the 4C-contact peak showed a 3-fold increase in luciferase activity.

To assess whether the deletion in the 4C-contact peak affected the expression levels of *ITPR1*, we measured *ITPR1* expression in the blood of the patient carrying the deletion and three unrelated, apparently healthy individuals. The relative *ITPR1* expression levels varied markedly between the three healthy individuals. While the patient showed an *ITPR1* expression similar to Control 1, they showed a 4–5-fold decrease in expression compared to controls 2 and 3 (Figure 5 and Appendix A).

## 3. Discussion

This is the first study to experimentally identify the enhancers of four SCA genes, *ATXN1*, *ATXN3*, *TBP* and *ITPR1,* in the human cerebellum in order to broaden our knowledge of the molecular mechanisms regulating the expression of these genes and assist in the discovery of novel genetic variants causing SCA. Using a comprehensive strategy involving 4C-seq on post-mortem human cerebellum tissue, annotation and filtering using publicly available datasets regarding the human cerebellum, as well as reciprocal 4C-seq and functional validation using luciferase assays, we identified two specific enhancers for each of the four SCA genes in the human cerebellum. In our undiagnosed patient cohort, we did not identify a CNV in any of the active enhancers.

*ITPR1* is a dosage-sensitive gene for which haploinsufficiency is known to cause SCA15 [17]. This means that there may be unidentified genetic variants within the enhancers of *ITPR1* that cause lower expression levels, resulting in SCA15. We found two active enhancers for *ITPR1*, R-PutE2, and R-PutE7, and the latter is located in the enhancer-associated non-coding RNA *BHLHE40-AS1* [19]. Reciprocal 4C-seq analysis of the other PutEs of *ITPR1* (PutE1, PutE3, PutE4, PutE5 and PutE6) did not confirm their direct interaction with the *ITPR1* promoter but did reveal a cluster of mutually interacting PutEs and R-PutEs. While enhanced luciferase activity demonstrated that PutE1, PutE5, and PutE6 are active enhancers, the significance of their interactions with the promoter of *ITPR1* was not confirmed by our reciprocal 4C-seq experiments. However, reciprocal 4C-seq for PutE1, PutE5, and PutE6 showed significant 4C-contact peaks close to the *ITPR1* promoter (Appendix A), suggesting that these PutEs might be engaged in a more transient interaction with the target promoter [20,21]. The CNV deletion we identified in one patient overlapped with a 4C-contact peak of *ITPR1* but not with PutE1, which was delineated in this peak based on public datasets. However, the absence of the deleted region in our in-house control population and the potential enhancer activity confirmed with the luciferase assay led us to study the functional consequence of the deletion in the patient. A study performed by Di Gregorio et al. showed that the reduced expression of *ITPR1* in blood can be used as a marker to identify SCA15 patients [22]. In our study, the *ITPR1* expression level in the patient (70-year-old female) was similar to that in one of the healthy individuals (64-year-old male) but much lower than that in two other healthy individuals (36-year-old male and 43-year-old female). The inconsistency in the *ITPR1* expression levels among the healthy individuals makes it difficult to draw a conclusion about the causality of this deletion. Future studies that utilize healthy individuals matched to the patient for sex and age as the controls, along with familial segregation studies, may help reveal the significance of this deletion. Furthermore, we suggest that future studies screen the active enhancers of *ITPR1*, as well as *ATXN1*, *ATXN3* and *TBP*, for other types of variants such as smaller deletions, insertions, and single-nucleotide variants. This would be particularly interesting for *ATXN3* since there is evidence that certain single-nucleotide variants in the 3′UTR are associated with early onset SCA3 [23]. Furthermore, current evidence suggests that disruption of TADs may interfere with gene expression, potentially leading to disease phenotypes. Therefore, we also screened for CNVs that could have altered TAD structures and contributed to the disease; however, no such variations were identified.

It has already been shown that, in addition to the expanded polyglutamine tract in ATXN1 protein that causes SCA1, slight increases in wild-type ATXN1 levels can also cause SCA1 in mice [13]. Studies suggest targeting the expression levels of *ATXN1* at the RNA and protein levels for therapeutic purposes in SCA1 patients [24,25]. While mouse studies have already shown that the 3′UTR and 5′UTR of *ATXN1* are crucial for *ATXN1* regulation via miRNAs2 [13,26], enhancers of *ATXN1* have not yet been studied. Other studies similarly support that reducing the levels of *ATXN3* mutant transcript/protein is a promising preventive treatment strategy [14,27]. An expanded polyQ tract in *ATXN3* causes SCA3, the most common type of SCA, a devastating disease for which there is currently no preventive treatment [14]. Studies have shown that ATXN3 is a non-essential protein and that reducing the expression level of both the normal and disease-causing allele will probably have tolerable side effects [27]. In mice, it has already been shown that inducing artificial miRNAs expressing RNA-interfering sequences that target the *ATXN3* 3′UTR can suppress the expression of *ATXN3* transcripts, an approach that can be used as a preventive treatment for SCA3 [27]. In this study, we identified two active enhancers for *ATXN1* (R-PutE1 and R-PutE2) and two for *ATXN3* (R-PutE1 and R-PutE2). Of all the R-PutEs tested in this study, *ATXN3* R-PutE1 showed the highest fold increase in the luciferase activity (a 59.4-fold increase), making it the strongest active enhancer detected in our study. Thus, the active enhancers identified for *ATXN1* and *ATXN3* may also be used as potential targets to reduce the expression of these genes, providing novel treatment strategies for SCA1 and SCA3 patients. A new class of drugs, bromo- and extra-terminal (BET) inhibitors, that target enhancers to inhibit gene expression are currently under investigation for the treatment of disorders linked to impaired enhancer function [28].

The general assumption is that most enhancer–promoter pairs reside in the same TAD [29] and that most of the active enhancers we identified for *ATXN1*, *ATXN3*, *TBP*, and *ITPR1* are indeed in the same TAD as the corresponding promoter. However, we also identified active enhancers not located in the same TAD as the promoter, including *ATXN1* R-PutE1 and *ATXN1* promoter, which were located in two neighboring TADs, as well as *ITPR1* R-PutE7 and *ITPR1* promoter, which were also located in two different TADs. While the majority of regulatory loops are a result of intra-TAD looping, regulatory looping across TADs (inter-TAD looping) has also been reported [18]. The boundary of the two TADs in which R-PutE1 and the *ATXN1* promoter are located appears to be quite weak, and the interactions between the two TADs (inter-TAD) can explain the contact between R-PutE1 and the *ATXN1* promoter. While the boundary of the two TADs containing R-PutE7 and the *ITPR1* promoter appears to be strong based on the Hi-C data, resulting in two separate TADs, our 4C-seq data suggest a very strong interaction between the two TADs.

The disease mechanism of SCA17 is a sequesteration of wild-type TBP in polyQ-pathological TBP aggregates in a dominant-negative manner, causing the loss of wild-type TBP. TBP is an important general transcription factor that also plays a role in the mechanism of other polyQ disorders (SCA1, SCA2, SCA3, Huntington’s disease, and dentatorubral-pallidoluysian atrophy) by sequestering in the polyQ-pathological aggregates of those diseases [15,16]. The loss of TBP has a serious impact on the highly regulated process of DNA transcription, where any minor disruption could lead to major consequences for the cells [15,30]. The two active enhancers of *TBP* that we identified, R-PutE1 and R-PutE2, encompass *DYNLT2* and *LINC00242/LINC00574*, respectively. During transcriptional activation, long intergenic non-protein-coding RNAs (LINCRNAs) are transcribed from their genes and can act as enhancer-RNAs (eRNAs) [31,32]. Chromatin looping between the genes coding for LINCRNAs and the target promoter brings the transcribed LINCRNAs eRNAs near the target gene promoter to allow for gene activation [31]. The reciprocal 4C-seq data of R-PutE1 (encompassing *DYNLT2*) and R-PutE2 (encompassing *LINC00574*) revealed a mutual interaction between these two regions (Appendix A), suggesting that *LINC00574* has a regulatory effect on *DYNLT2*. Surprisingly, in a recent study, *LINC00574* was suggested to regulate the expression of the nearby gene *DYNLT2* (*TCTE3*) [33]. Thus, our results not only show that *LINC00574* regulates the expression of *TBP* but also support *LINC00574*’s regulatory role for *DYNLT2.* The enhancers of *TBP* we identified will provide insight into the molecular mechanisms governing the expression of this tightly regulated gene in polyQ-mediated neurodegenerative disorders.

Our study has several limitations. First, it is known that enhancers can be more than a Mb away from their target promoters [34]. However, the signal of the 4C-seq technique is highest and most reproducible within ~500 kb of the viewpoint, limiting 4C-seq’s ability to detect weaker/more transient interactions at larger distances [35]. We could therefore have missed relevant enhancers in these more distant regions. Such enhancers would be detectable using extremely deep-sequenced Hi-C or Micro-C, two techniques that can identify interactions with increased signals over backgrounds at any length-scale [36,37]. Second, the activity of the enhancers was assessed with luciferase assays, which have their own drawbacks. For example, transcription factors present in the correct cell type and genomic context necessary for the proper functioning of the enhancers might be missing in this in vitro system, although we attempted to mimic the environment by using the most appropriate cell line available. Moreover, all the enhancers were tested with an identical vector carrying a minimal promoter, but this can be a problem for enhancers that are only functional with their own target promoters. Third, we cannot conclude with complete certainty that the identified active enhancers are also functional for their target genes in the cerebellum, as enhancers can be active at different stages and timepoints. Fourth, the absence of CNVs in the identified enhancers among the patients may be due to the low genotyping resolution of the SNP array, which could have missed smaller CNVs in the active enhancers. Supporting this, a previous study we conducted demonstrated that CNVs detected with the limited resolution of the SNP array are a rare cause of SCA [38]. It is also possible that other types of genetic variations more commonly associated with SCAs—such as deletions, insertions, or single-nucleotide variations in the identified enhancers—are the cause of the disease in the patients but were not screened in our study. Lastly, it could be that the limitations of our enhancer detection approach caused us to miss a key enhancer harboring disease-causing variations. Fifth, the lack of clinical severity at a cohort level is a limitation, as it could have provided insights into disease variability and helped guide future genetic analyses.

A more direct way to study the impact of the identified active enhancers on *ATXN1*, *ATXN3*, *TBP*, and *ITPR1* expression would be to knock-out the active enhancers with CRISPR/Cas9 in a proper cell line (e.g., SHSY5Y cells) and assess the target gene expression. Moreover, testing the functionality of the active enhancers in zebra fish can provide further insight into tissue specificity and in vivo activity patterns during development [7,39]. Finally, as we only searched for CNVs in the active enhancers, we strongly suggest screening these regions for other types of variations in genetically undiagnosed patients in order to discover novel genetic variants.

## 4. Materials and Methods

To identify enhancers of *ATXN1*, *ATXN3*, *TBP,* and *ITPR1* in the human cerebellum, we used the following workflow (Figure 6): (I) circularized chromosome conformation capture sequencing (4C-seq) on the human cerebellum to capture genomic regions interacting with the promoters of the four SCA genes; (II) peak-calling to identify *cis*-genomic regions that interact significantly with gene promoters (4C-contact peaks); (III) annotation of these 4C-contact peaks using publicly available datasets to identify putative enhancers (PutEs); (IV) reciprocal 4C-seq for the PutEs to confirm interactions with the associated gene promoters (R-PutEs); (V) analysis of the enhancer activity of R-PutEs using luciferase assays to identify active enhancers. Following the identification of the enhancers, we screened these regions for copy number variants (CNVs) in a cohort of genetically undiagnosed SCA patients.

### 4.1. 4C-Sequencing

**Promoter viewpoint selection.** We obtained the promoter regions and transcription start sites (TSSs) of *ATXN1*, *ATXN3*, *TBP*, and *ITPR1* from the Eukaryotic promoter database (https://epd.expasy.org/epd/ (accessed on 1 February 2018)) and the database of TSS (https://dbtss.hgc.jp/ (accessed on 1 January 2018)), respectively. Promoter regions selected as viewpoints included the TSS (Appendix A).

**4C template preparation.** Cerebellar tissue from three healthy individuals was obtained from the Netherlands Brain Bank (http://www.brainbank.nl/ (accessed on 1 March 2018)) and stored at −80 °C. For each 4C-template, 100 mg of frozen tissue was dissected at −20 °C and homogenized on ice in pre-cooled 2 mL PBS with 10% Fetal Bovine Serum (FBS) using a cold 15 mL Dounce homogenizer. The cell suspensions obtained from the cerebellar tissue were immediately processed using the previously published 4C protocol [40]. In brief, after cross-linking the cells with 2% formaldehyde, they were lysed, and the DNA was digested with 400 U of NlaIII restriction enzyme (RE) (New England Biolabs, Ipswich, MA, USA), followed by ligation with 50 U of T4 DNA ligase (Merck, Darmstadt, Germany). Following de-crosslinking of the ligated DNA concatemers with proteinase K (Thermo Fisher Scientific, Waltham, MA, USA), the DNA was purified with NucleoMag P-beads (Macherey-Nagel, Düren, Germany). A second round of digestion was performed with 50 U of DpnII or Csp6I RE (Thermo Fisher Scientific), followed by ligation with 100 U T4 DNA ligase and a purification step with NucleoMag P-beads. Two 4C-templates were generated using the REs. For each 4C-template, three replicates were generated using human cerebellum samples from the three individuals. The 4C template used for each promoter viewpoint can be found in Appendix A.

**Preparation of 4C sequencing libraries.** Primer design and sequencing library preparation were performed as described in Krijger et al. [40] with some adjustments to the library preparation step. For each viewpoint of interest, specific reading and non-reading primers were designed for the first PCR to amplify genomic regions interacting with the viewpoints of interest (Appendix A). Per viewpoint, this PCR was performed in eight reactions of 25 µL (200 µL total) using 150 ng of 4C-template per reaction and the Expand Long Template PCR System (Roche, Basel, Switzerland) (Appendix A). After pooling the PCR products, AMPure XP beads (Beckman Coulter, Brea, CA, USA) were used to purify the products. A second round of PCR was performed on the initial PCR products (Appendix A) with universal primers containing Illumina adaptor sequences and indexes (Appendix A). AmPure XP beads were used to purify the final PCR products. The PCR products were analyzed with the Agilent D5000 Screen Tape System (Agilent Technologies, Santa Clara, CA, USA) to measure the fragment size and DNA concentrations in the range of 250–5000 bp.

**Sequencing.** We pooled 18 to 24 4C-seq libraries obtained from different viewpoints/replicates. The pools were loaded conservatively (0.9 pM) and included 25% PhiX to ensure optimal diversity for base calling. Sequencing was performed on an Illumina Nextseq500 (Illumina, San Diego, CA, USA) using the Mid-Output v2 kit (Illumina, San Diego, CA, USA) with a single-read run and 75 bp read length.

**Data processing and 4C contact peak-calling.** Sequenced libraries were first demultiplexed based on the reading primers to separate the reads of each gene. We then used Illumina indexes introduced by the second PCR to separate the replicates of each gene. Demultiplexed data were analyzed using the 4C-seq pipeline of Krijger et al. [40]. In brief, reads were trimmed based on the first RE site in order to extract the captured sequence that also carries the RE motif. The trimmed reads were then mapped to the human reference genome (hg19) using Bowtie2. To count reads that map to the RE fragment ends, the reference genome was in silico digested and the fragment ends were mapped against the genome to identify fragment ends that were unique to different RE combinations. After smoothing and normalizing the read counts, the data were further analyzed in R version 4.2.2. using the PeakC peak-calling algorithm to identify genomic regions with statistically significant interactions with the viewpoints of interest known as 4C-contact peaks [41]. Peak-calling was based on three biological replicates with parameters set on alphaFDR = 0.05, qWr = 1, and qWd = 1.5.

### 4.2. Identification of PutEs

To identify the PutEs of *ATXN1*, *ATXN3*, *TBP* and *ITPR1* in the human cerebellum, the 4C-seq contact peaks were intersected and annotated using publicly available datasets in ENCODE (https://www.encodeproject.org/ (accessed on 1 June 2020)) and GEO (https://www.ncbi.nlm.nih.gov/geo/ (accessed on 1 June 2020)). For the human cerebellar tissue, ATAC-seq (GSE101912) [42], ChIP-seq for H3K27ac (GSM1119154) and DNase-seq (GSM736538) data were used. Hi-C data (GSM2824367) from the cerebellum were used for the analysis of topologically associated domains (TADs). The PyGenome track was used to analyze and visualize the genome tracks [43]. For gene annotation, we used the Genome Browser annotation track database to export data from the following: group: genes and gene predictions, track: UCSC genes and table: known genes (https://genome.ucsc.edu/cgi-bin/hgTables (accessed on 1 June 2020)). PutEs for each gene were identified as the section of the 4C-contact peak in which the chromatin was open (based on the ATAC-seq and DNase-seq data) and carried the H3K27ac active enhancer marker. Hi-C data were used to identify TADs and study the location of the promoters and PutEs with respect to TADs. The data analysis and visualization procedures can be found at https://github.com/GhorbaniF/4CSeqEnhancerDetection (accessed on 1 June 2020).

### 4.3. Reciprocal 4C-Seq

Reciprocal 4C-seq was performed for each PutE viewpoint (Appendix A) to identify the genomic regions interacting with the PutEs and confirm their interaction with the original gene promoters. The 4C-template used for the PutE viewpoints and the specific reading and non-reading primers can be found in Appendix A. We used the same 4C-sequencing procedure described earlier to perform the reciprocal 4C-seq experiments. To identify regions significantly interacting with the PutE viewpoints (4C-contact peaks), we performed peak-calling with two biological replicates and alphaFDR = 0.2, qWr = 1, and qWd = 1.5. Upon the original promoter being identified as one of the 4C-contact peaks of the PutE, we considered the interaction between the promoter and PutE to be confirmed, and these PutEs were designated as Reciprocal PutEs (R-PutEs). The reciprocal 4C-seq data of the PutEs was also used to reveal interactions between the PutEs/R-PutEs identified for each gene.

### 4.4. Luciferase Assays

Candidate regions were PCR-amplified using human genomic DNA (Roche, Basel, Switzerland) as a template and cloned in pGL4.23 firefly luciferase reporter vector (Promega, Madison, WI, USA). In addition, a previously reported validated enhancer region was PCR-amplified and cloned as a positive control [7]. Since it has been demonstrated that the sizes of transfected constructs are inversely correlated with luciferase activity [44], for the negative control, we cloned a 1479 bp stuffer DNA into the pGL4.23 firefly luciferase reporter vector in order to ensure similar insert sizes for the test constructs (which ranged from 768 to 2690 bp) and the negative control construct (see primer sequences in Appendix A). Candidate regions > 3 kb were amplified as two overlapping segments (the size of the overlap was, on average, 1800 bp) and cloned into two separate pGL4.23 firefly luciferase reporter vectors to facilitate the amplification and cloning procedures. Constructs were transfected into SH-SY5Y cells because this cell line is a widely used model for neurodegenerative disorders [45]. The pGL4.23 constructs were co-transfected with a Renilla internal control vector pRL-TK (Promega, Wisconsin, WI, USA) in a 50:1 ratio in SH-SY5Y cells in 6-well plates with a seeding density of 1 million cells per well. Cells were grown in Dulbecco’s modified Eagle medium supplemented with 15% FBS and 1% penicillin-streptomycin in a 37 °C incubator with 5% CO_2_. Transfections were performed in triplicate using nucleofection (Lonza, Thermo Fisher Scientific, Waltham, MA, USA) according to the manufacturer’s instructions. Luciferase assays were performed 48 h after transfection using the Dual-Luciferase Reporter Assay System (Promega) following the manufacturer’s instructions. The activity of the firefly and Renilla for each sample were measured with a Promega GloMax^®^ 96 Microplate Luminometer (Promega). Per well, the firefly luciferase activity was normalized by dividing the firefly signal by the Renilla signal. The fold change in the normalized firefly luciferase activity for each candidate region is expressed as follows:$$\mathrm{Fold}\;\mathrm{change}=\frac{Normalized\;firefly\;luciferase\;activity\;of\;the\;candidate\;region}{Normalized\;firefly\;luciferase\;activity\;of\;the\;negative\;control}$$

Candidate enhancer regions with at least a two-fold increase in luciferase activity were regarded as active enhancers.

### 4.5. CNV Analysis in SCA Patients Using SNP Array

**Patient selection**. We studied 260 patients with sporadic or familial adult-onset cerebellar ataxias who were referred to the Department of Genetics at the University Medical Center Groningen in the Netherlands for SCA genetic diagnostics. The cohort consisted of 149 males and 111 females with a mean age at onset of 60.1 years. At the time of inclusion, none of the patients had received a genetic diagnosis after testing for coding repeat expansions in *ATXN1-3*, *CACNA1A*, *ATXN7*, and *TBP*, as well as screening for single-nucleotide variants and small insertions or deletions in the conventional SCA genes. Experiments were performed in all samples in accordance with the regulations and ethical guidelines of the University Medical Center of Groningen (PaNaMaID 7679). All patients gave informed consent for use of diagnostically obtained materials for innovations of diagnostic care.

**Array analysis**. A genome-wide SNP array was performed using the Infinium Global Screening Array-24 v3.0-EA-MD (Illumina) following the manufacturer’s protocol. Data processing was conducted with in-house pipelines (https://github.com/molgenis/AGCT (accessed on 1 September 2021) and https://github.com/molgenis/GAP (accessed on 1 September 2021)). Processed data were uploaded into NxClinical version 5.0 software (BioDiscovery, El Segundo, CA, USA) to perform CNV analysis and visualize CNVs in the active enhancers of *ATXN1*, *ATXN3*, *TBP*, and *ITPR1*. To exclude artifacts, only CNVs comprising at least 10 subsequent SNPs were selected for further analysis. An in-house genome-wide CNV database comprising data from 3280 healthy Dutch controls was used to filter out benign variants. Sanger sequencing was performed to confirm the break points of the identified CNVs (for primer sequences see Appendix A).

### 4.6. Gene Expression Analysis of ITPR1

To assess the effect of a CNV deletion found near a potential *ITPR1* enhancer region in one patient, we collected blood from the patient in a PAXgene tube (PreAnalytiX, Hombrechtikon, Switzerland) for mRNA isolation. For controls, we obtained whole-blood RNA samples from three apparently healthy unrelated individuals of different ages: a 64-year-old male, a 36-year-old male, and a 43-year-old female (Amsbio, Cambridge, MA, USA). mRNA was reverse-transcribed using the RevertAid H Minus First Strand cDNA Synthesis Kit (Thermo Fisher Scientific). To measure *ITPR1* expression levels in blood, a quantitative RT-PCR analysis was performed using iTaq Universal SYBR^®^ Green Supermix (Thermo Fisher Scientific) on a Thermo Fisher Quantstudio 7 flex Momentum machine according to the manufacturer’s instructions. The geomean of two housekeeping genes, *GAPDH* and *Beta-actin*, was used for normalization of raw *ITPR1* expression values. Two sets of primers were used to measure *ITPR1* expression levels, and one set was used for *GAPDH* and *Beta-actin* (Appendix A). We used the delta–delta Ct method (2^–∆∆Ct^ method) to calculate the relative fold change in gene expression. The relative gene expression for each individual is based on three independent replicates.

A summary table listing all genes mentioned in this study, along with their chromosomal positions and known functions, is included in Appendix A.

## 5. Conclusions

In conclusion, we identified several active enhancers for *ATXN1*, *ATXN3*, *TBP*, and *ITPR1*. The output of this study provides insights into the regulation of the genes in the human cerebellum, which can lead to the discovery of novel genetic variants causing SCA and to new therapeutic options for SCA patients. The cohort used in our study is relatively small, and the low resolution of the SNP array may have resulted in missed CNVs in the enhancer regions. Therefore, screening a larger SCA cohort using a higher-resolution approach for CNV detection might still uncover interesting findings in our regions of interest. Moreover, the identification of these elements would provide a starting point to study other types of variations in these regions. Since it is being increasingly recognized that genetic variations that disturb enhancer functionality can cause brain disorders, we strongly suggest identifying and screening the enhancers of other SCA genes, in particular for those genes with known variants within the regulatory regions (*DAB1*, *PPP2R2B*, and *ATXN8OS*), to explain a fraction of missing genetic diagnoses for SCA patients.

## Figures and Tables

**Figure 1 ijms-25-11205-f001:**
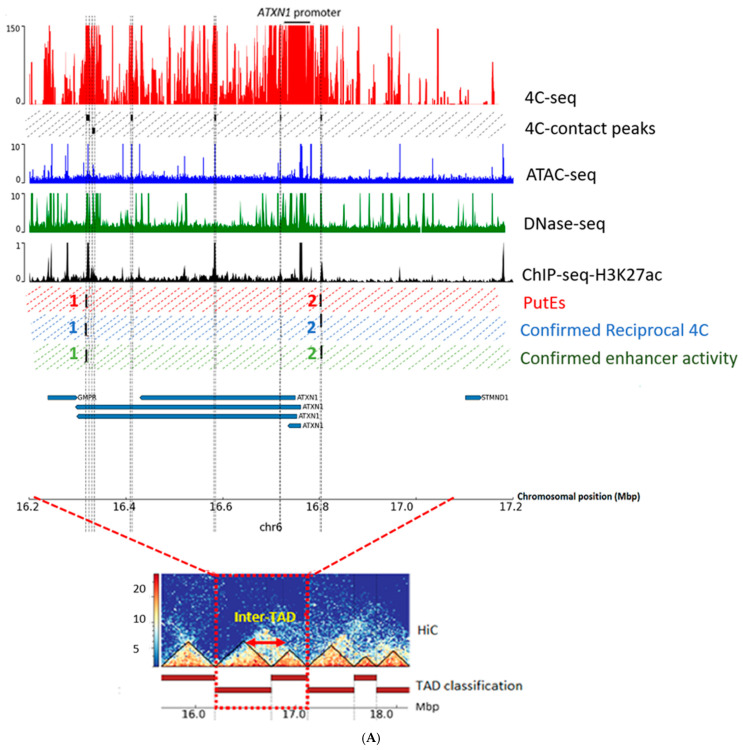
Identification of enhancers of (**A**) *ATXN1*, (**B**) *ATXN3*, (**C**) *TBP*, and (**D**) *ITPR1* in the human cerebellum. The red track shows the 4C-seq data generated for the promoter of the corresponding gene. Black boxes show genomic regions significantly interacting with the promoters (4C-contact peaks). Blue, green, and black tracks show publicly available human cerebellum datasets for ATAC-seq, Dnase-seq, and ChIP-seq for H3K27ac, respectively. Red, blue, and green dashed boxes represent, respectively, putative enhancers (PutEs), PutEs confirmed to interact with their associated promoter through reciprocal 4C-seq (R-PutEs), and R-PutEs confirmed to have enhancer activity. The bottom track shows the genomic coordinates and the genes (UCSC gene track; hg19). TADs obtained from the human cerebellum HiC data are depicted in the bottom plot.

**Figure 2 ijms-25-11205-f002:**
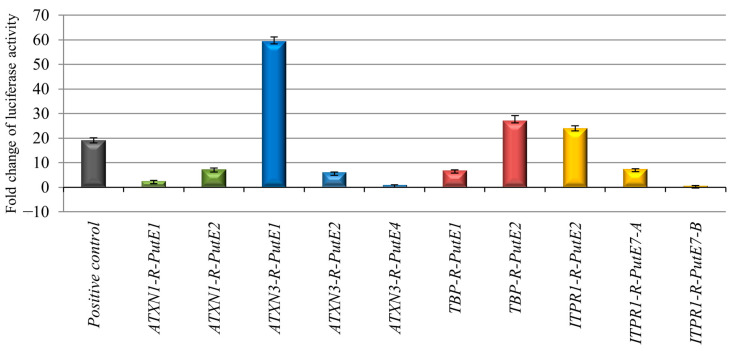
Fold change in the luciferase activity of the R-PutEs of *ATXN1*, *ATXN3*, *TBP*, and *ITPR1*. Fold change is expressed as the ratio of the normalized luciferase activity (firefly/Renilla) of the candidate region to that of the negative control (pGL4.23 firefly luciferase reporter vector with a 1479 bp stuffer DNA). A previously reported validated enhancer was used as a positive control. Fold change values are based on the means of three independent experiments. Standard deviations are shown as error bars. Each color represents the R-PutEs of a gene.

**Figure 3 ijms-25-11205-f003:**
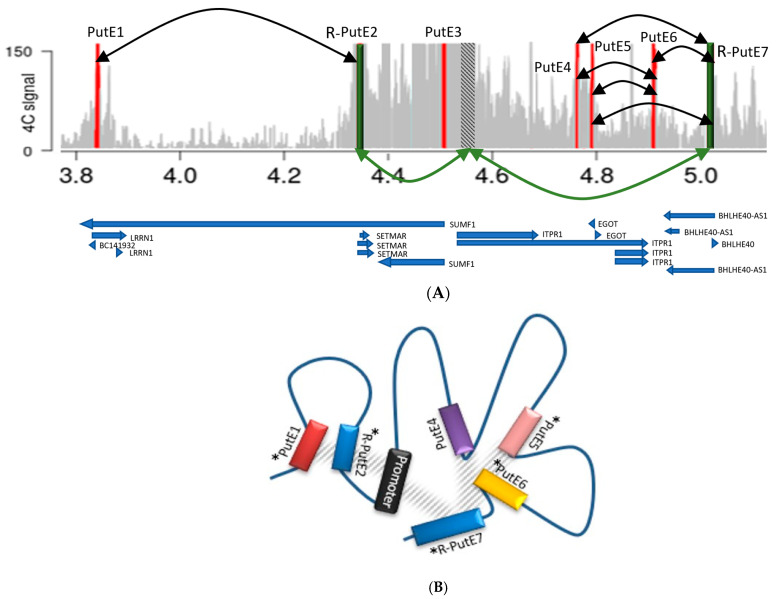
Analysis of the PutE–R-PutE interactions for *ITPR1*. (**A**) 4C-seq and reciprocal 4C-seq data for the *ITPR1* promoter and PutEs revealed mutual interactions between PutEs and R-PutEs. Red bars: PutEs. Green bars: R-PutEs. Dashed bar: *ITPR1* promoter. Green arrows: mutual interaction between the *ITPR1* promoter and R-PutEs. Black arrows: mutual interactions between PutEs and R-PutEs. Chromosomal locations and gene annotations are indicated underneath the figure. (**B**) A schematic representation of the hypothetical interactions between *ITPR1* PutEs and R-PutEs. Only R-PutE2 and R-PutE7 mutually interact with the *ITPR1* promoter. Reciprocal 4C-seq revealed significant interactions between PutE1, R-PutE2, PutE4, PutE5, PutE6, and R-PutE7. Asterisks indicate PutEs and R-PutEs which showed an increase in luciferase activity. Mutual interactions between PutEs and R-PutEs are shown as gray hatching. (**C**) Fold change in the luciferase activity of PutE1, R-PutE2, PutE4, PutE5, PutE6, and R-PutE7. Fold change is expressed as the ratio of normalized luciferase activity (firefly/Renilla) of the candidate region to that of the negative control (pGL4.23 firefly luciferase reporter vector with a 1479 bp stuffer DNA). A previously reported validated enhancer was used as a positive control. Fold change values are based on the means of three independent experiments. Standard deviations are shown as error bars. Each color represents one of the R-PutEs/PutEs of the *ITPR1* gene.

**Figure 4 ijms-25-11205-f004:**
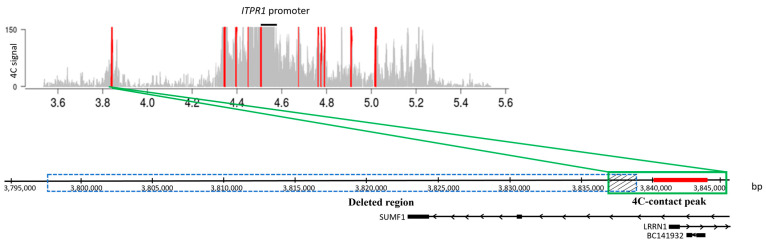
Schematic representation of the deleted region in one patient and its overlap with the 4C-contact peak of *ITPR1*. The upper plot shows the significant 4C-contact peaks (red bars) of the *ITPR1* promoter. The lower plot shows the deleted region in the patient (GRCh37: chr3:3,797,792–3,838,896; blue dotted box), which partly overlaps (dashed box) the broad 4C-contact peak (GRCh37: chr3:3,836,985–3,845,608; green box) of the *ITPR1* promoter. PutE1 (red line in the lower plot), the active enhancer region delineated in the 4C-contact peak, does not overlap with the deleted region in the patient. Gene annotations are indicated underneath the figure. The deletion encompasses part of intron 11 from the old version of the *SUMF1* transcript (NM_182760) but does not encompass the updated transcript (NM_182760.4).

**Figure 5 ijms-25-11205-f005:**
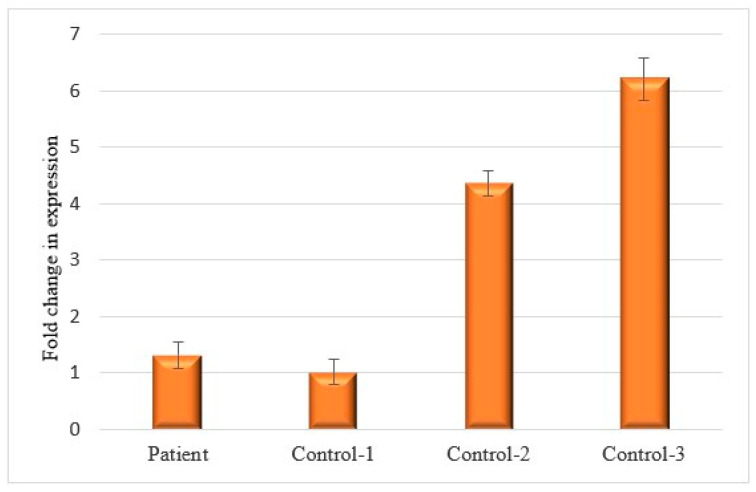
Expression of *ITPR1* in the blood of an SCA patient carrying a CNV deletion in a 4C-contact peak of the *ITPR1* promoter compared to expression in three healthy individuals. The delta–delta Ct was used to determine the relative fold change in expression levels. Expression values are normalized to Control 1. The relative fold change in expression for each individual was based on the means of three independent replicates. Standard deviations are shown as error bars. Patient: 70-year-old female, Control 1: 64-year-old male, Control 2: 36-year-old male, Control 3: 43-year-old female.

**Figure 6 ijms-25-11205-f006:**
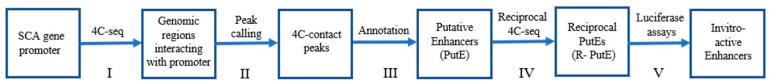
Workflow used to identify enhancers of *ATXN1*, *ATXN3*, *TBP* and *ITPR1* in the human cerebellum.

## Data Availability

The data used to support the findings of this study are available from the corresponding author upon request.

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
