# Peer review of "Identification and Copy Number Variant Analysis of Enhancer Regions of Genes Causing Spinocerebellar Ataxia"

_ijms, 2024, doi:10.3390/ijms252011205_

Round 1
Reviewer 1 Report
Comments and Suggestions for Authors
In this manuscript the authors make efforts to identify enhancers of 4 genes known to cause spinocerebellar ataxia. They were able to identify several potential enhancers for all of these genes. Because in about 40% of SCA cases no sequence changes in the known SCA genes are found, it is likely that in many of these cases mutations or copy number variants might be present in the enhancers of such genes. The authors tried to identify such changes in their newly identified enhancer regions but failed to find such changes in a cohort of 285 patients without mutations in the known SCA genes. In one patient a deletion in the ITPR1 enhancer region was found, but it could not be confirmed that this deletion caused the disease phenotype.
This is an interesting manuscript which for the first time identifies enhancer regions of genes known to cause SCA. While the methods for identifying enhancers are complicated and difficult to perform the authors convincingly show that the regions identified by their assays are likely enhancers of the relevant genes. This is an important achievement because it will allow to expand genetic testing to these regions.
Interestingly, the authors did not find mutations, deletions or duplications in these regions in their cohort of 285 patients with SCA of unclear origin. Unfortunately, this finding is only poorly discussed in the manuscript. The authors should discuss this negative finding more extensively and weigh arguments that this is due to the fact that the concept that many cases of SCA are caused by mutations in the enhancer regions is wrong (e.g. because of a substantial redundancy of the enhancer elements) or whether it is due to rather incomplete identification of the enhancers by their approach. As the authors have identified the relevant TADs for the different enhancers, they may also have the possibility to look for sequence abnormalities in the entire TADs rather than only in the enhancer regions of the undiagnosed patient cohort. It was not clear to me whether this approach was already undertaken in the manuscript or whether the study only focused on the identified enhancer regions.
In summary, this is an interesting manuscript which for the first time identifies enhancers for some of the most common SCA genes. There is some room for improvement of the discussion which avoids the issue of why no changes were found in the undiagnosed SCA cohort.
Reviewer 2 Report
Comments and Suggestions for Authors
The manuscript “Identification and copy number variant analysis of enhances regions of genes causing spinocerebellar ataxia” presents a novel attempt to pathogenesis of some of the neurodegenerative disorders.
The paper is very interesting. It is well written and all the steps of identification and searching for enhancers in the 4 genes are precisely described. The section Materials and Methods is well prepared. The section Discussion is quite long, but I feel it is necessary concerning a lot of elements confirming the validity of the study. The limitations are concerned and discussed, pointing to develop the more extensive research in this field.
A minor comment concerns:
- I am not sure if the results of the controls (healthy individuals) were presented?
Summing up, the manuscript gives a lot of novel and potentially useful information on functional activity of the ataxia-associated genes and the other studies may embrace other genes in the future.
Reviewer 3 Report
Comments and Suggestions for Authors
It is strictly fundamental to present the 4 SCA genes in the introduction, not in the discussion!
It is necessary to insert a summary table of a lot of genes mentioned in the paper, with the chromosomal position and function!
Line 171-174: can you give some detalis on the vectors used?
The big limitation of this paper is that you do not present at all the cohort of 285 patients: you must at least distinguish them by sex, race, age of onset, clinical severity. You must also include any genetic screening performed.
Line 216: you write about three healthy unrelated individuals of different age, and you do not write if they are matched for sex or age. Which age did you choose?
The same is in lines 405-406 where you do not explain the differencess between these healthy individuals (how did you choose these?)
Line 502-503: correct, because you write about our SCA cohort (that you did not define) and other SCA cohorts (different from what point of view? Genetic, age, race?)
Comments on the Quality of English Language
Lines 42-44, lines 49-51, lines 89-90, lines 159-161, line 482 (did try), line 41 (demonstrateS) are of poor English quality: can you correct?
Line 177: renilla or Renilla?
Reviewer 4 Report
Comments and Suggestions for Authors
This manuscript is well presented and with an important goal, that is to intercept the potential etiology of a set of patients with spinocerebellar ataxia by defining yet unknow enhancer regions in a subset of known SCA genes. The authors identified several active enhancers for ATXN1, ATXN3, TBP and ITPR1. They also offered insights into the regulation of the genes in the human cerebellum, which can lead to the discovery of new therapeutic options for SCA patients.
Searching for CNVs in the specific enhancer regions, the authors did not find any pathological genomic rearrangement able to explain the disease status.
The study is sound, and limitations are well presented and results properly balanced.
Minor modifications
The term 285 patients with cerebellar ataxia with no etiology is too general and the authors should indicate a) the clinical status (cerebellar ataxia, sensory ataxia, spinocerebellar ataxia, spastic ataxia), b) age at onset and disease duration, c) pattern of transmission (sporadic, autosomal dominant o recessive or X-linked) to allow a more precise definition of their studied cohort.
Round 2
Reviewer 3 Report
Comments and Suggestions for Authors
Unfortunately, they do not have data on clinical severity at a cohort level, and I think it is strictly necessary to underline this big limitation of the study.
Reviewer 4 Report
Comments and Suggestions for Authors
no comments
Author Response
Thank you for considering our responses to the comments. We appreciate your time and feedback.